# Dataset Distillation

## Abstract

Model distillation aims to distill the knowledge of a complex model into a simpler one. In this paper, we consider an alternative formulation called *dataset distillation*: we keep the model fixed and instead attempt to distill the knowledge from a large training dataset into a small one. The idea is to *synthesize* a small number of data points that do not need to come from the correct data distribution, but will, when given to the learning algorithm as training data, approximate the model trained on the original data. For example, we show that it is possible to compress $60,000$ MNIST training images into just $10$ synthetic *distilled images* (one per class) and achieve close to original performance with only a few steps of gradient descent, given a particular fixed network initialization. We evaluate our method in a wide range of initialization settings and with different learning objectives. Experiments on multiple datasets show the advantage of our approach compared to alternative methods in most settings.

## 1 Introduction

Hinton et al. (2015) proposed *network distillation* as a way to transfer the knowledge from an ensemble of many separately-trained networks into a single, typically compact network, performing a type of model compression. In this paper, we are considering a related but orthogonal task: rather than distilling the model, we propose to distill the dataset. Unlike network distillation, we keep the model fixed but encapsulate the knowledge of the entire training dataset, which typically contains thousands to millions of images, into a small number of *synthetic* training images. In fact, we show that we can go as low as *one* synthetic image per category, training the same model to reach surprisingly good performance on these synthetic images. For example in Fig. 1a, we compress $60,000$ training images of MNIST digit dataset into only $10$ synthetic images (one per class), given a fixed network initialization. Training the standard LENET (LeCun et al., 1998) architecture on these $10$ images yields test-time MNIST recognition performance of $94\%$, compared to $99\%$ for the original task. For networks with unknown random weights, $100$ synthetic images train to $80\%$ with a few gradient descent steps. We name our method *Dataset Distillation* and these images *distilled images*.

But why is dataset distillation useful? There is the purely scientific question of how much data is really encoded in a given training set and how compressible it is? Moreover, given a few distilled images, we can now "load up" a given network with an entire dataset-worth of knowledge much more efficiently, compared to traditional training that often uses tens of thousands of gradient descent steps.

A key question is whether it is even possible to compress a dataset into a small set of synthetic data samples. For example, is it possible to train an image classification model on synthetic images that are not on the natural image manifold? Conventional wisdom would suggest that the answer is no, as the synthetic training data may not follow the same distribution as the real test data. Yet, in this work, we show that this is indeed possible. We present a new optimization algorithm for synthesizing a small number of synthetic data samples not only capturing much of the original training data but also tailored explicitly for fast model training in only a few gradient steps. To achieve our goal, we first derive the network weights as a differentiable function of our synthetic training data. Given this connection, instead of optimizing the network weights for a particular training objective, we can optimize the pixel values of our distilled images. However, this formulation requires access to the initial network weights of the network. To relax this assumption, we develop a method for generating distilled images for networks with random initializations from a certain distribution. To further boost performance, we propose an iterative version, where we obtain a sequence of distilled images to train a model and each distilled image can be trained with multiple passes. Finally, we study the case of a

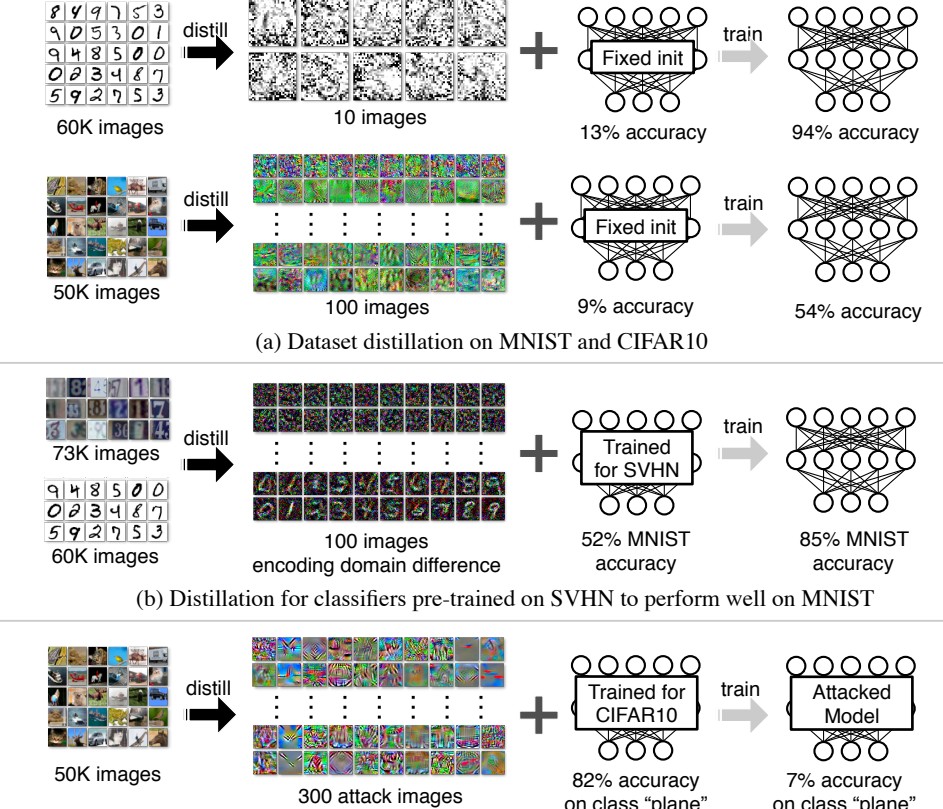

(a) Dataset distillation on MNIST and CIFAR10

(b) Distillation for classifiers pre-trained on SVHN to perform well on MNIST

(c) Distillation for a malicious objective on well-trained CIFAR10 classifiers

Figure 1: Dataset Distillation: we distill the knowledge of tens of thousands of images into a few synthetic training images called distilled images. (a): On MNIST, 10 distilled images can train a standard LENET with a particular fixed initialization to 94% test accuracy (compared to 99% when fully trained). On CIFAR10, 100 distilled images can train a deep network with fixed initialization to 54% test accuracy (compared to 80% when fully trained). (b): Using pre-trained networks for SVHN, we can distill the domain difference between two SVHN and MNIST into 100 distilled images. These images can be used to quickly fine-tune networks trained for SVHN to achieve high accuracy on MNIST. (c): Training for a malicious objective, our formulation can be used to create adversarial attack images. If well-optimized networks retrained with these images for **one single** gradient step, they will catastrophically misclassify a particular targeted class.

simple linear model, deriving a lower bound on the size of distilled data required to achieve the same performance as training on the full dataset.

We demonstrate that a handful of distilled images can be used to train a model with a fixed initialization to achieve surprisingly high performance. For a network with unknown random weights pre-trained on other tasks, our method can still find distilled images for fast model fine-tuning. We further test our method on a wide range of initialization settings: fixed initialization, random initialization, fixed pre-trained weights, and random pre-trained weights, as well as two training objectives: image classification and malicious dataset poisoning attack. Extensive experiments on four publicly available datasets, MNIST (LeCun, 1998), CIFAR10 (Krizhevsky & Hinton, 2009), PASCAL-VOC (Everingham et al., 2010) and CUB-200 (Wah et al., 2011), show that our method often performs better than alternative methods and existing baselines. Our code and models will be available upon publication.

## 2 RELATED WORK

**Knowledge distillation.** The main inspiration for this paper is network distillation (Hinton et al., 2015), a widely used technique in ensemble learning (Radosavovic et al., 2018) and model compression (Ba & Caruana, 2014; Romero et al., 2015; Howard et al., 2017). While network distillation aims to distill the knowledge of multiple networks into a single model, our goal is to compress the knowledge of an entire dataset into a few synthetic training images. Our method is also related to the theoretical concept of teaching dimension, which specifies the size of dataset necessary to teach a

target model (oracle) to a learner (Goldman & Kearns, 1995; Shinohara & Miyano, 1991). While these methods do not enforce the training data to be real, they need the existence of oracle models, which our method does not require.

**Dataset pruning, core-set construction, and instance selection.** Another way to distill knowledge is to summarize the entire dataset by a small subset, either by only using the "valuable" data for model training (Angelova et al., 2005; Lapedriza et al., 2013; Felzenszwalb et al., 2010) or by only labeling the "valuable" data via active learning (Cohn et al., 1996; Tong & Koller, 2001). Similarly, core-set construction (Bachem et al., 2017; Tsang et al., 2005; Har-Peled & Kushal, 2007; Sener & Savarese, 2018) and instance selection (Olvera-López et al., 2010) methods aim to select a subset of the entire training data, such that models trained on the subset will perform as closely well as possible to the model trained on full dataset for faster training time. For example, solutions to many classical linear learning algorithms, e.g., Perceptron (Rosenblatt, 1957) and support vector machine (SVMs) (Hearst et al., 1998), are weighted sums of a subset of training examples, which can be viewed as core-sets. However, algorithms constructing these subsets require many more training examples per category than we do, in part because their "valuable" images have to be real, whereas our distilled images are exempt from this constraint.

**Gradient-based hyperparameter optimization.** Our work bears similarity with the gradient-based hyperparameter optimization techniques, which compute the gradient of hyperparameter w.r.t. the final validation loss by reversing the entire training procedure (Bengio, 2000; Domke, 2012; Pedregosa, 2016; Maclaurin et al., 2015). We also backpropagate errors through optimization steps. However, we use only training set data and focus much more heavily on learning synthetic training data rather than tuning hyperparameters. To our knowledge, this direction has only been slightly touched on previously (Maclaurin et al., 2015). We explore it in much greater depth and demonstrate the idea of dataset distillation through various settings. More crucially, our distilled images can work well across random initialization weights, which cannot be achieved by any prior work.

**Understanding datasets.** Researchers have presented various approaches for understanding and visualizing learned models (Zeiler & Fergus, 2014; Zhou et al., 2015; Mahendran & Vedaldi, 2015; Bau et al., 2017; Koh & Liang, 2017). Unlike these approaches, we are interested in understanding the intrinsic properties of the training data rather than a specific trained model. Analyzing training datasets has, in the past, been mainly focused on the investigation of bias in datasets (Ponce et al., 2006; Torralba & Efros, 2011). For example, Torralba & Efros (2011) proposed to quantify the "value" of dataset samples using cross-dataset generalization. Our method offers a new perspective for understanding datasets by distilling full datasets into few synthetic samples.

## 3    APPROACH

Given a model and a dataset, we aim to obtain a new, much-reduced *synthetic* dataset which performs almost as well as the original dataset. We first present our main optimization algorithm for training a network with a fixed initialization with one gradient descent (GD) step (Sec. 3.1). In Sec. 3.2, we derive the resolution to a more challenging case, where the initial weight is random rather than fixed. We also discuss the initial weights distribution where our method can work well. Furthermore, we study a linear network case to help the readers understand both the solution and limits of our method in Sec. 3.3. In Sec. 3.4, we extend our approach to more than one gradient descent steps and more than one passes. Finally, Sec. 3.5 and Sec. 3.6 demonstrate how to obtain distilled images with different initialization distributions and learning objectives.

Consider a training dataset $\mathbf{x} = \{x_i\}_{i=1}^N$. We parameterize our neural network as $\theta$ and denote $\ell(x_i, \theta)$ as the loss function that represents the loss of this network on a data point $x_i$. Our task is to find the minimizer of the empirical error over the entire training data:

$$\theta^* = \arg\min_\theta \frac{1}{N} \sum_{i=1}^N \ell(x_i, \theta) = \arg\min_\theta \ell(\mathbf{x}, \theta), \tag{1}$$

where for notation simplicity we overload the $\ell(\cdot)$ notation so that $\ell(\mathbf{x}, \theta)$ represents the average error of $\theta$ over the entire dataset $\mathbf{x} = \{x_i\}_{i=1}^N$. We make the mild assumption that $\ell$ is twice-differentiable, which holds for the majority of modern machine learning models (e.g., most neural networks) and tasks.

---

**Algorithm 1** Dataset Distillation

---

**Input:** $p(\theta_0)$: distribution of initial weights; $M$: the number of distilled data
**Input:** $\alpha$: step size; $n$: batch size; $T$: the number of optimization iterations; $\tilde{\eta}_0$: initial value for $\tilde{\eta}$
1: Initialize $\tilde{\mathbf{x}} = \{\tilde{x}_i\}_{i=1}^M$ randomly, $\tilde{\eta} \leftarrow \tilde{\eta}_0$
2: **for each** training step $t = 1$ to $T$ **do**
3:     Get a minibatch of real data $\mathbf{x}_t = \{x_{t,j}\}_{j=1}^n$
4:     Sample a batch of initial weights $\theta_0^{(j)} \sim p(\theta_0)$
5:     **for each** sampled $\theta_0^{(j)}$ **do**
6:         Compute updated parameter with GD: $\theta_1^{(j)} = \theta_0^{(j)} - \tilde{\eta}\nabla_{\theta_0^{(j)}}\ell(\tilde{\mathbf{x}}, \theta_0^{(j)})$
7:         Evaluate the objective function on real data: $\mathcal{L}^{(j)} = \ell(\mathbf{x}_t, \theta_1^{(j)})$
8:     **end for**
9:     Update $\tilde{\mathbf{x}} \leftarrow \tilde{\mathbf{x}} - \alpha\nabla_{\tilde{\mathbf{x}}}\sum_j \mathcal{L}^{(j)}$, and $\tilde{\eta} \leftarrow \tilde{\eta} - \alpha\nabla_{\tilde{\eta}}\sum_j \mathcal{L}^{(j)}$
10: **end for**
**Output:** distilled data $\tilde{\mathbf{x}}$ and the optimized learning rate $\tilde{\eta}$

---

### 3.1 OPTIMIZING DISTILLED DATA

Standard training usually applies minibatch stochastic gradient descent (SGD) or its variants. At each step $t$, we sample a minibatch of training data $\mathbf{x}_t = \{x_{t,j}\}_{j=1}^n$ and update the current parameters as

$$\theta_{t+1} = \theta_t - \eta\nabla_{\theta_t}\ell(\mathbf{x}_t, \theta_t),$$

where $\eta$ is the learning rate. Such a training process often takes tens of thousands or even millions of above update steps to converge. Instead, we aim to learn a tiny set of synthetic distilled training data $\tilde{\mathbf{x}} = \{\tilde{x}_i\}_{i=1}^M$ with $M \ll N$ and a corresponding learning rate $\tilde{\eta}$ so that a single GD step like

$$\theta_1 = \theta_0 - \tilde{\eta}\nabla_{\theta_0}\ell(\tilde{\mathbf{x}}, \theta_0) \tag{2}$$

using these learned synthetic data $\tilde{\mathbf{x}}$ greatly boosts performance on the real training dataset.

Given an initialization $\theta_0$, we obtain these synthetic data and $\tilde{\eta}$ that minimize the objective below $\mathcal{L}$:

$$\tilde{\mathbf{x}}^*, \tilde{\eta}^* = \underset{\tilde{\mathbf{x}},\tilde{\eta}}{\arg\min}\,\mathcal{L}(\tilde{\mathbf{x}}, \tilde{\eta}; \theta_0) = \underset{\tilde{\mathbf{x}},\tilde{\eta}}{\arg\min}\,\ell(\mathbf{x}, \theta_1) = \underset{\tilde{\mathbf{x}},\tilde{\eta}}{\arg\min}\,\ell(\mathbf{x}, \theta_0 - \tilde{\eta}\nabla_{\theta_0}\ell(\tilde{\mathbf{x}}, \theta_0)), \tag{3}$$

where we derive the new weights $\theta_1$ as a function of distilled images $\tilde{\mathbf{x}}$ and learning rate $\tilde{\eta}$ using Eqn. 2 and then evaluate the new weights over all the training images $\mathbf{x}$. Note that the loss $\mathcal{L}(\tilde{\mathbf{x}}, \tilde{\eta}; \theta_0)$ is differentiable w.r.t. $\tilde{\mathbf{x}}$ and $\tilde{\eta}$, and can thus be optimized using standard gradient-based algorithms. In many classification tasks, the data $\mathbf{x}$ may contain discrete parts, e.g., the class labels in data-label pairs. For such cases, we fix the discrete parts rather than learn them.

### 3.2 DISTILLATION FOR RANDOM INITIALIZATIONS

Unfortunately, the above distilled data optimized for a given initialization do not generalize well to other initialization weights. The distilled data often look like random noise (e.g., in Fig. 2a) as it encodes the information of both training dataset $\mathbf{x}$ and a particular network initialization $\theta_0$. To address the above issue, we turn to calculate a small number of distilled data that can work for networks with random initializations from a specific distribution. We formulate the optimization problem as follows:

$$\tilde{\mathbf{x}}^*, \tilde{\eta}^* = \underset{\tilde{\mathbf{x}},\tilde{\eta}}{\arg\min}\,\mathbb{E}_{\theta_0 \sim p(\theta_0)}\mathcal{L}(\tilde{\mathbf{x}}, \tilde{\eta}; \theta_0), \tag{4}$$

where $\theta_0$ is a randomly sampled network initialization from the distribution $p(\theta_0)$. Algorithm 1 illustrates our main method. During optimization, the distilled data are optimized to work well for multiple networks whose initial weights are sampled from $p(\theta_0)$. In practice, we observe that the final distilled data generalize well to the unseen initializations. Besides, these distilled images usually look quite informative, encoding the discriminative features of each category (Fig. 3).

For distilled data to be properly learned, it turns out to be crucial for $\ell(\mathbf{x}, \cdot)$ to share similar local conditions (e.g., output values, gradient magnitudes) over $\theta_0$ sampled from $p(\theta_0)$. In the next section, we derive a lower bound on the number of distilled data needed for a simple model with arbitrary initial $\theta_0$, and discuss its implications on choosing $p(\theta_0)$.

## 3.3 Analysis of a Simple Linear Case with Quadratic Loss

This section studies our formulation in a simple linear regression case. We derive the lower bound of the number of distilled images needed to achieve the same performance as training on full dataset for arbitrary initialization with one GD step. Consider a dataset $\mathbf{x}$ containing $N$ data-target pairs $\{(d_i, t_i)\}_{i=1}^{N}$, where $d_i \in \mathbb{R}^D$ and $t_i \in \mathbb{R}$, which we represent as two matrices: an $N \times D$ data matrix $\mathbf{d}$ and an $N \times 1$ target matrix $\mathbf{t}$. Given the mean squared error and a $D \times 1$ weight matrix $\theta$, we have

$$\ell(\mathbf{x}, \theta) = \ell((\mathbf{d}, \mathbf{t}), \theta) = \frac{1}{2N} \|\mathbf{d}\theta - \mathbf{t}\|^2. \tag{5}$$

We aim to learn $M$ synthetic data-target pairs $\tilde{\mathbf{x}} = (\tilde{\mathbf{d}}, \tilde{\mathbf{t}})$, where $\tilde{\mathbf{d}}$ is an $M \times D$ matrix, $\tilde{\mathbf{t}}$ an $M \times 1$ matrix ($M \ll N$), and $\tilde{\eta}$ the learning rate, to minimize $\ell(\mathbf{x}, \theta_0 - \tilde{\eta}\nabla_{\theta_0}\ell(\tilde{\mathbf{x}}, \theta_0))$. The updated weight matrix after one GD step with these distilled data is

$$\theta_1 = \theta_0 - \tilde{\eta}\nabla_{\theta_0}\ell(\tilde{\mathbf{x}}, \theta_0) = \theta_0 - \frac{\tilde{\eta}}{M}\tilde{\mathbf{d}}^T(\tilde{\mathbf{d}}\theta_0 - \tilde{\mathbf{t}}) = (\mathbf{I} - \frac{\tilde{\eta}}{M}\tilde{\mathbf{d}}^T\tilde{\mathbf{d}})\theta_0 + \frac{\tilde{\eta}}{M}\tilde{\mathbf{d}}^T\tilde{\mathbf{t}}. \tag{6}$$

Note that for such quadratic loss, there always exists some learned distilled data $\tilde{\mathbf{x}}$ allowing us to achieve the same performance as training on full dataset $\mathbf{x}$ (i.e., attaining the global minimum) for *any* initialization $\theta_0$.[*] But how small can $M$, the size of distilled data, be? For such models, the global minimum is attained at any $\theta^*$ satisfying $\mathbf{d}^T\mathbf{d}\theta^* = \mathbf{d}^T\mathbf{t}$. Substituting Eqn. (6) in, we have

$$\mathbf{d}^T\mathbf{d}(\mathbf{I} - \frac{\tilde{\eta}}{M}\tilde{\mathbf{d}}^T\tilde{\mathbf{d}})\theta_0 + \frac{\tilde{\eta}}{M}\mathbf{d}^T\mathbf{d}\tilde{\mathbf{d}}^T\tilde{\mathbf{t}} = \mathbf{d}^T\mathbf{t}. \tag{7}$$

Here we make the mild assumption that the feature columns of the data matrix $\mathbf{d}$ are independent (i.e., $\mathbf{d}^T\mathbf{d}$ has full rank). For a $\tilde{\mathbf{x}} = (\tilde{\mathbf{d}}, \tilde{\mathbf{t}})$ to satisfy the above equation for any $\theta_0$, we must have

$$\mathbf{I} - \frac{\tilde{\eta}}{M}\tilde{\mathbf{d}}^T\tilde{\mathbf{d}} = \mathbf{0}, \tag{8}$$

which implies that $\tilde{\mathbf{d}}^T\tilde{\mathbf{d}}$ has full rank and $M \geq D$.

**Discussion.** The analysis considers only a simple case but suggests that any small number of distilled data fails to generalize to arbitrary starting $\theta_0$. This is intuitively expected as the optimization target $\ell(\mathbf{x}, \theta_1) = \ell(\mathbf{x}, \theta_0 - \tilde{\eta}\nabla_{\theta_0}\ell(\tilde{\mathbf{x}}, \theta_0))$ depends on the local behavior of $\ell(\mathbf{x}, \cdot)$ around $\theta_0$, which can be drastically different across various $\theta_0$ values. We note that the lower bound $M \geq D$ is a quite restricting one, considering that real datasets often have thousands to even hundreds of thousands of dimensions (e.g., image classification). This analysis motivates us to focus on $p(\theta_0)$ distributions that yield similar local conditions over the support. Sec. 3.5 discusses several practical choices explored in this paper. Additionally, to address the limitation of using a single GD step, we extend our method to multiple GD steps in the next section. In Sec. 4.1, we empirically verify that using multiple steps is much more effective than using just one on deep convolutional networks, with the total amount of distilled data fixed.

## 3.4 Multiple Gradient Descent Steps and Multiple Epochs

We can extend Algorithm 1 to more than one gradient descent steps by changing Line 6 to multiple sequential GD steps each on a different batch of distilled data and learning rate, i.e., each step $i$ is

$$\theta_{i+1} = \theta_i - \tilde{\eta}_i\nabla_{\theta_i}\ell(\tilde{\mathbf{x}}_i, \theta_i), \tag{9}$$

and changing Line 9 to backpropagate through all steps. However, naively computing gradients is both memory-intensive and computationally-expensive. Therefore, we exploit a recent technique called *back-gradient optimization*, which allows for significantly faster gradient calculation of such updates in reverse-mode differentiation (i.e., backpropagation). Specifically, back-gradient optimization formulates the necessary second order terms into efficient Hessian-vector products (Pearlmutter, 1994), which can be easily calculated with modern automatic differentiation systems such as PyTorch (Paszke et al., 2017). For further algorithm details in this aspect, we refer readers to prior work (Domke, 2012; Maclaurin et al., 2015).

**Multiple epochs.** To further improve the performance, we can train the network with the same distilled images for multiple epochs (passes) of the GD step(s). In particular, we tie the image pixels

---

[*]One choice is to pick any global minimum $\theta^*$, and choose $\tilde{\mathbf{d}} = N \cdot \mathbf{I}$ and $\tilde{\mathbf{t}} = N \cdot \theta^*$.

for the same distilled images used in different epochs. In other words, for each epoch, our method cycles through all GD steps, where each step is associated with a different batch of distilled data. We do not tie the trained learning rates across epochs as later epochs often use smaller learning rates.

## 3.5 DISTILLATION WITH DIFFERENT INITIALIZATIONS

Inspired by the analysis of the simple linear case in Sec. 3.3, we aim to focus on initial weights distributions $p(\theta)$ that yield similar local conditions over the support. In this work, we focus on the following four practical choices:

- **Random initialization:** Distribution over model weights initialized using methods that attempts to ensure gradient flow of constant magnitude, e.g., He Initialization (He et al., 2015) and Xavier Initialization (Glorot & Bengio, 2010) for convolutional neural networks (CNNs).

- **Fixed initialization:** A fixed initial weights sampled using the method above.

- **Random pre-trained weights:** Distribution over models pre-trained on other tasks and datasets, e.g., pre-trained ALEXNET (Krizhevsky et al., 2012) networks for ImageNet classification (Deng et al., 2009). Each network is pre-trained on the same task, but with different initializations.

- **Fixed pre-trained weights:** A fixed model weights pre-trained on other tasks and datasets.

**Distillation for pre-trained weights.** Such learned distilled data essentially fine-tunes weights pre-trained on one task to perform well for a new task, thus bridging the gap between two domains. Domain mismatch and dataset bias represent a challenging problem in machine learning today (Torralba & Efros, 2011). Extensive prior work has been proposed to adapt models to new tasks and datasets (Daume III, 2007; Saenko et al., 2010). In this work, we characterize the domain mismatch via distilled data. In Sec. 4.2, we show that a very small number of distilled images are sufficient to quickly adapt CNN models to new classification tasks.

## 3.6 DISTILLATION WITH DIFFERENT OBJECTIVES

Previous sections show that we can train distilled data to minimize the loss of the distilled task $\ell(\mathbf{x}, \theta_1)$ defined on the final updated weights $\theta_1$ (Line 7 in Algorithm 1). Distilled images trained with different final learning objectives can train models to exhibit different desired behaviours. We have already mentioned image classification as one of the applications, where distilled images help train accurate classifiers. Below, we introduce a quite different training objective to further demonstrate the flexibility of our method.

**Distillation for a malicious data-poisoning objective.** For example, our approach can be used to construct a new form of data poisoning attack. To illustrate this idea, we consider the following scenario. When a single GD step is applied with our synthetic adversarial data, a well-behaved image classifier catastrophically forgets a category but still maintains high performance on other categories.

Formally, given an attacked category $K$ and a target category $T$, we want the classifier to misclassify images from category $K$ to category $T$. To achieve this, we consider a new final objective function $\ell_{K \to T}(\mathbf{x}, \theta_1)$, which is a classification loss encouraging $\theta_1$ to classify category $K$ images mistakenly as category $T$ while correctly predicting other images, e.g., a cross entropy loss with target labels of $K$ modified to $T$. Then, the attacking distilled images can be obtained via optimizing

$$\tilde{\mathbf{x}}^*, \tilde{\eta}^* = \operatorname*{arg\,min}_{\tilde{\mathbf{x}}, \tilde{\eta}} \mathbb{E}_{\theta_0 \sim p(\theta_0)} \mathcal{L}_{K \to T}(\tilde{\mathbf{x}}, \tilde{\eta}; \theta_0) = \operatorname*{arg\,min}_{\tilde{\mathbf{x}}, \tilde{\eta}} \mathbb{E}_{\theta_0 \sim p(\theta_0)} \ell_{K \to T}(\mathbf{x}, \theta_1), \qquad (10)$$

where $p(\theta_0)$ is the distribution over *random pre-trained weights* of well-optimized classifiers.

Compared to prior data poisoning attacks (Biggio et al., 2012; Li et al., 2016; Muñoz-González et al., 2017; Koh & Liang, 2017), our approach crucially *does not* require the poisoned training data to be stored and trained on repeatedly. Instead, our method attacks the model training just in one iteration and with only a few data. This advantage makes our method effective for many online training algorithms and useful for the case where malicious users hijack the data feeding pipeline for only one gradient step (e.g., one network transmission). In Sec. 4.2, we show that a single batch of distilled data applied in one step can successfully attack well-optimized neural network models. This setting can be viewed as distilling dataset knowledge of a specific category into data.

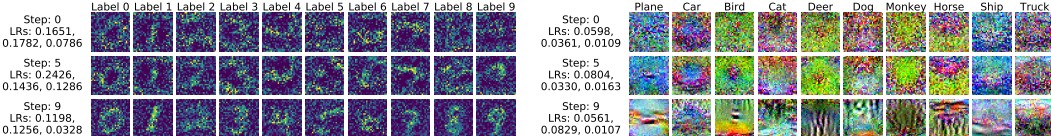

(a) MNIST. These images train networks with a partic- (b) CIFAR10. These images train networks with a partic-
ular initialization from 12.9% test accuracy to 93.76%. ular initialization from 8.82% test accuracy to 54.03%.

Figure 2: Distilled images trained for *fixed initialization*. MNIST distilled images use 1 GD step and 3 epochs (10 images in total). CIFAR10 distilled images use 10 GD steps and 3 epochs (100 images in total). For CIFAR10, only selected steps are shown. At left, we report the corresponding learning rates for all 3 epochs.

(a) MNIST. These images train networks with unknown (b) CIFAR10. These images train networks with un-
initialization to $79.50\% \pm 8.08\%$ test accuracy. known initialization to $36.79\% \pm 1.18\%$ test accuracy.

Figure 3: Distilled images trained for *random initialization* with 10 GD steps and 3 epochs. We show images from selected GD steps and corresponding trained learning rates for all 3 epochs.

## 4 EXPERIMENTS

We report image classification results on MNIST (LeCun, 1998) and CIFAR10 (Krizhevsky & Hinton, 2009). For MNIST, distilled images are trained with LENET (LeCun et al., 1998), which achieves about 99% test accuracy if fully trained. For CIFAR10, we use a network architecture following Krizhevsky (2012) which achieves around 80% test accuracy if fully trained. For random initializations and random pre-trained weights, we report means and standard deviations on 200 held-out models, unless otherwise specified.

**Baselines.** For each experiment, in addition to baselines specific to the setting, we generally compare our method against baselines trained with data derived or selected from real images:

- **Random real images:** We randomly sample the same number of real training images per category.
- **Optimized real images:** We sample sets of real images as above, and choose on the top 20% sets that perform the best training images.
- **$k$-means:** For each category, we use $k$-means to extract the same number of cluster centroids as the number of distilled images in our method.
- **Average real images:** We compute the average image of all the images in each category, which is reused in different GD steps.

For these baselines, we perform each evaluation on 200 hold-out models with all combinations of learning rate $\in \{$learned learning rate with our method$, 0.001, 0.003, 0.01, 0.03, 0.1, 0.3\}$ and #epochs $\in \{1, 3, 5\}$. We report results from the best performing combination. We run all the experiments on NVIDIA Titan Xp and V100 GPUs. We use one GPU for fixed initial weights and four GPUs for random initial weights. Each training typically takes 1 to 4 hours. Please see supplemental material Sec. S-6.1 for more training and baseline details.

### 4.1 DATASET DISTILLATION

**Fixed initialization.** With access to initial network weights, distilled images can directly train a particular network to reach high performance. For example, 10 learned distilled images can boost the test accuracy of a neural network with an initial accuracy 12.90% to the final accuracy 93.76% on MNIST (Fig. 2a). Similarly, 100 images can train a network with an initial accuracy 8.82% to 54.03% test accuracy on CIFAR10 (Fig. 2b). This result suggests that even only a few distilled images have enough capacity to distill part of the dataset.

**Random initialization.** Trained with randomly sampled initializations using Xavier initialization (Glorot & Bengio, 2010), the learned distilled images do not need to encode information tailored for a particular starting point and thus can represent meaningful content independent of network initializations. In Fig. 3, we see that such distilled images reveal discriminative features of the corresponding categories: e.g., the ship image in Fig. 3b. These 100 images can train randomly

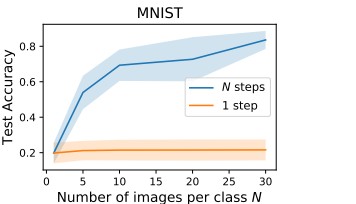

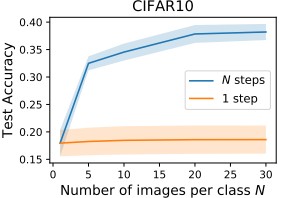

(a) Effect of number of steps        (b) Effect of number of epochs

Figure 4: Hyperparameter sensitivity studies on *random initialization*: (a) average test accuracy w.r.t. the number of gradient descent steps. The number of epochs is fixed to be 2. (b) average test accuracy w.r.t. the number of epochs. The number of steps is fixed to be 10, with each containing 10 images (one per category).

Figure 5: Comparison between applying the same number of images in one versus multiple GD steps on *random initialization*, with the number of epochs fixed to 1. $N$ denotes the total number of images per category. For multiple steps runs, each of the $N$ steps applies one image per category.

| | Ours | | Baselines | | | | | |
|---|---|---|---|---|---|---|---|---|
| | Fixed init. | Random init. | Used as training data in same number of GD steps | | | | Used as data for K-NN classification | |
| | | | Random real | Optimized real | $k$-means | Average real | Random real | $k$-means |
| MNIST | **96.6%** | 79.5% $\pm$ 8.1% | 68.6% $\pm$ 9.8% | 73.0% $\pm$ 7.6% | 76.4% $\pm$ 9.5% | 77.1% $\pm$ 2.7% | 71.5% $\pm$ 2.1% | **92.2%** $\pm$ **0.1%** |
| CIFAR10 | **54.0%** | **36.8%** $\pm$ **1.2%** | 21.3% $\pm$ 1.5% | 23.4% $\pm$ 1.3% | 22.5% $\pm$ 3.1% | 22.3% $\pm$ 0.7% | 18.8% $\pm$ 1.3% | 29.4% $\pm$ 0.3% |

Table 1: Comparison between our method trained for 10 GD steps and 3 epochs and various baselines. For baselines using K-Nearest Neighbor (K-NN), best result among all combinations of distance metric $\in \{l_1, l_2\}$ and $\mathrm{K} \in \{1, 3\}$ is reported. In K-NN and $k$-means, $\mathrm{K}$ and $k$ can have different values. All methods use 10 images per class, except for the average real images baseline, which reuses the same images in different GD steps.

initialized networks to 36.79% average test accuracy on CIFAR10. Similarly, for MNIST, the 100 distilled images shown in Fig. 3a can train randomly initialized networks to 79.50% test accuracy.

**Multiple gradient descent steps and multiple epochs.** In Fig. 3, we learn distilled images for 10 GD steps applied in 3 epochs, leading to a total of 100 images (with each step containing one image per category). In each epoch, these 10 steps are sequentially applied once. The early steps tend to look noisier, likely regularizing random weights to point easier for further optimization. In later steps, the images gradually look like real data and share the discriminative features for these categories. Fig. 4a shows that using more steps significantly improves the results. Fig. 4b shows a similar but slower trend as the number of epochs increases. We observe that longer training (i.e., more epochs) can help the model learn all the knowledge from the distilled images, but the performance is eventually limited by the capacity of the images (i.e., the number of total images). Alternatively, we can train the model with one GD step but a big batch size. Sec. 3.3 has shown theoretical limitations of using only one step in a simple linear case. In Fig. 5, we empirically verify that with convolutional networks, using multiple steps drastically outperforms single step method, with the same number of distilled images.

Table 1 compares our method against several baselines. Our method with both fixed and random initialization outperform all the baselines on CIFAR10 and most of the baselines on MNIST.

## 4.2 DISTILLATION FOR DIFFERENT INITIALIZATIONS AND OBJECTIVES

Next, we show two extended settings of our main algorithm discussed in Sec. 3.5 and Sec. 3.6. Both cases assume that the initial weights are random but pre-trained on a different dataset. We train the distilled images on 2000 random pre-trained models, and then apply them on *unseen* models.

**Fixed and random pre-trained weights on digits.** As shown in Sec. 3.5, we can optimize distilled images to quickly fine-tune pre-trained models for a new dataset. Table 2 shows that our method is more effective compared to various baseline on adaptation among three digits datasets: MNIST, USPS (Hull, 1994), and SVHN (Netzer et al., 2011). We also compared against a state-of-the-art few-short supervised domain adaptation method (Motiian et al., 2017). Although our method uses the entire training set to compute the distilled images, both methods use the same number of images to distill the knowledge of target dataset. Prior work (Motiian et al., 2017) is outperformed by our method with

(a) Accuracy w.r.t. incorrect labels      (b) Ratio of attacked category misclassified as target

Figure 6: Performance for our method and baselines with *random pre-trained initialization and a malicious objective*. Distilled images are trained for one GD step. For baselines, we use the same numbers of images with incorrect labels and also apply one GD step, and report the result that achieves the highest accuracy w.r.t. the incorrect labels while having $\geq 10\%$ misclassification ratio on the attacked category, to avoid results with learning rates too low to change model behavior at all. (a) Our method slightly outperforms the best baseline in accuracy w.r.t. incorrect labels. (b) Our method performs similarly with some baselines in changing the prediction of the attacked category on MNIST, but is much better than all baselines on CIFAR10.

| | Ours with fixed pre-trained | Ours with random pre-trained | Random real | Optimized real | $k$-means | Average real | Domain adaptation Motiian et al. (2017) | No adaptation | Train on **full** destination training set |
|---|---|---|---|---|---|---|---|---|---|
| $\mathcal{M} \to \mathcal{U}$ | **97.9**% | $95.4\% \pm 1.8\%$ | $94.9\% \pm 0.8\%$ | $95.2\% \pm 0.7\%$ | $92.2\% \pm 1.6\%$ | $93.9\% \pm 0.8\%$ | $\mathbf{96.7\% \pm 0.5\%}$ | $90.4\% \pm 3.0\%$ | $97.3\% \pm 0.3\%$ |
| $\mathcal{U} \to \mathcal{M}$ | **93.2**% | $92.7\% \pm 1.4\%$ | $87.1\% \pm 2.9\%$ | $87.6\% \pm 2.1\%$ | $85.6\% \pm 3.1\%$ | $78.4\% \pm 5.0\%$ | $89.2\% \pm 2.4\%$ | $67.5\% \pm 3.9\%$ | $98.6\% \pm 0.5\%$ |
| $\mathcal{S} \to \mathcal{M}$ | **96.2**% | $85.2\% \pm 4.7\%$ | $84.6\% \pm 2.1\%$ | $85.2\% \pm 1.2\%$ | $\mathbf{85.8\% \pm 1.2\%}$ | $74.9\% \pm 2.6\%$ | $74.0\% \pm 1.5\%$ | $51.6\% \pm 2.8\%$ | $98.6\% \pm 0.5\%$ |

Table 2: Performance of our method and baselines in adapting models among MNIST ($\mathcal{M}$), USPS ($\mathcal{U}$), and SVHN ($\mathcal{S}$). 100 distilled images are trained for 10 GD steps and 3 epochs. Few-shot domain adaptation method by Motiian et al. (2017) and baselines use the same numbers image per class.

| Destination dataset | Ours | Random real | Optimized real | Average real | Fine-tune on **full** destination training set |
|---|---|---|---|---|---|
| PASCAL-VOC | **70.75**% | $19.41\% \pm 3.73\%$ | $23.82\% \pm 3.66\%$ | $9.94\%$ | $75.57\% \pm 0.18\%$ |
| CUB-200 | **38.76**% | $7.11\% \pm 0.66\%$ | $7.23\% \pm 0.78\%$ | $2.88\%$ | $41.21\% \pm 0.51\%$ |

Table 3: Performance of our method and baselines in adapting an ALEXNET pre-trained on ImageNet to PASCAL-VOC and CUB-200. Only one distilled image per class are trained to be applied in 1 GD step repeated for 3 epochs. Our method significantly outperforms the baselines. Results are collected over 10 runs.

fixed pre-trained weights on all the tasks, and by our method with random pre-trained weights on two of the three tasks. This result shows that our distilled images indeed convey compressed information of the full dataset.

**Fixed pre-trained weights on ImageNet.** In Table 3, we adapt a widely-used ALEXNET model (Krizhevsky, 2014) pre-trained on ImageNet (Deng et al., 2009) to perform image classification on PASCAL-VOC (Everingham et al., 2010) and CUB-200 (Wah et al., 2011) datasets. Using only 1 distilled image per category, our method outperforms the baselines significantly. Our result is also comparable to the accuracy of fine-tuning on the full datasets which contain thousands of images.

**Random Pre-trained weights and a malicious data-poisoning objective.** Sec. 3.6 shows that our method can construct a new type of data poisoning, where the attacker can apply just one GD step with a few malicious data to manipulate a well-trained model. We train distilled images to make well-optimized neural networks to misclassify a particular attacked category as another target category *within only one GD step*. Our method requires *no* access to the exact weights of the model. In Fig. 6b, we evaluate our method on 200 held-out models, against various baselines using data derived from real images and incorrect labels. While some baselines perform similarly well as our method on MNIST, our method significantly outperforms all the baselines on CIFAR10.

## 5 DISCUSSION

In this paper, we present dataset distillation for compressing the knowledge of entire training data into a few synthetic training images. We can train a network to reach high performance with a small number of distilled images and several gradient descent steps. Finally, we demonstrate two applications including fast domain adaptation and effective data poisoning attack. In the future, we plan to extend our method to compress large-scale visual datasets such as ImageNet (Deng et al., 2009) and other types of data (e.g., audio and text). Also, our current method is sensitive to the initial weights distribution. We would like to investigate more on various initialization strategies, with which distilled images can work well.

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

## S-6 Supplementary Material

### S-6.1 Experiment Details

For the networks used in our experiments, we disable dropout layers due to the randomness and computational cost they introduce in distillation. Moreover, we initialize the distilled learning rates as 0.02 and use Adam optimizer (Kingma & Ba, 2014) with a learning rate of 0.001. For random initialization and random pre-trained weights, we sample 4 to 16 initial weights in each step.

Details of the baselines are listed below.

- **Random real images:** We randomly sample the same number of real training images per category. 10 such set of sampled images are evaluated.
- **Optimized real images:** We sampled 50 sets of real images using above procedure, and evaluate 10 sets that achieve best performance on 20 held-out models and 1024 training images.
- **$k$-means:** For each category, we use $k$-means to extract the same number of cluster centroids as the number of distilled images in our method. 10 such set of sampled images are evaluated.
- **Average real images:** We compute the average image of all the images in each category, which is reused in different GD steps. We evaluate the model only once because average images are deterministic.

