# OpenReview forum: "Dataset Distillation"
_ICLR.cc/2019/Conference_

### Official Review · AnonReviewer2 · 2018-10-17
**Interesting approach, not fully mature yet**

**Rating:** 5
**Confidence:** 4

**Review:**

The paper addresses the interesting problem of generating a small number of synthetic examples that can be used to train a classifier, replacing a larger dataset.

The paper is clearly written, the approach makes sense, and the experiments are interesting.
My major concerns are regarding to previous literature, analysis of the algorithm, and details of the experiment. Overall, I expect an ICLR paper to go deeper (rather than wide). I recommend presenting strong convincing evidence on one front.


Specific comments:
(1)  I'm missing analysis of the proposed procedure. It wasn't fully clear which Loss it minimizes and if it indeed guaranteed to converge to the minimum of that loss.

(2)  The topic of learning from few samples is presented as completely new. It is well known that for classical linear algorithms like the Perceptron and SVM, the weights are a weighted sum of (label-weighted) samples, hence by definition of these algorithm, there is a single sample that can be used to "train" the model in one step. I'd expect some discussion of how the proposed approach relate to these classical approaches.
There is also existing literature on a related problem of selecting samples (Teaching dimension Goldman&Kearns) that could be somewhat relevant here.

(3) Motivation. The paper provide several motivations for dataset distillation. I support the first motivation of scientific understanding what data is actually needed for a classifier, and this means that deeper analysis is needed. The practical motivations are less convincing, because (a) domain adaptation experiments are not compared with real baselines (b) robustness of poisoning with a single sample is not studied/discussed.

(4) experiments: The intro states that training with 10 images reaches 94% accuracy, but this does not seem consistent with the results in Table 1. The caption of figure 2 suggests that accuracy is between 12% and 94% which means the stated 94% is not representative or typical. Could you clarify?
For domain adaptation. The baseline (random images) are very weak, and still perform almost   comparably to the proposed approach. More robust experiments are needed here: stronger baselines, decent hyper-parameter search etc.

(5) Writing and exposition: The paper addresses two issues: (a) learning with few synthetic samples, and (b) learning with few gradient steps. The intro tends to mix the two, and it is not clear why learning with a single gradient step is important. I recommend to separate the two topics more clearly.

---

> ### Author Response · Authors · 2018-11-27
> **Re:  Interesting approach, not fully mature yet**
>
> Q: Consistency among intro, Table 1, and Figure 2.
> A: These numbers are in fact consistent. We clarify the differences here. Given a fixed initialization that achieves 12% *initial* test accuracy on MNIST, ten distilled images can boost this network to 94% *final* test accuracy. Both intro and Figure 2 describes this setting in a consistent way. In Figure 5 (Table 1 in the previous version), the distilled images are trained for random unknown initializations setting, which produces numbers different from the fixed initialization setting. We have updated the captions of Figure 2 and Figure 5 accordingly.
>
> Q: Stronger baselines and more robust experiments
> A: We have added comparisons against many baselines for all the settings in this revision, including several methods proposed by R3, and a SOTA few-shot domain adaptation method [1].  Our method beats baselines and the domain adaptation method [1] on most tasks and datasets. Additionally, we have added a new experiment on large-scale datasets, where we adapt a pretrained AlexNet model to PASCAL-VOC and CUB-200. Our method significantly outperforms all the baselines (47% better on PASCAL-VOC and 31% better on CUB-200). Please refer to at Sec. 4.1 and Sec. 4.2 for more details.
>
> Q: Related work on classical linear algorithms and teaching dimension
> A: For algorithms whose solution is a weighted sum of training samples, such samples essentially form a core-set. Our works allow for the distilled data to be synthetic, while these prior work are limited to use real samples. Thank you for your reference on teaching dimension. It is a very relevant concept. However, while teaching dimension allows the training data to be synthetic, it needs an oracle model, which we do not require. In Sec. 2 of this revision, we discussed the classical linear algorithms in the new section on core-set construction and instance selection, and discussed teaching dimension in the section on knowledge distillation.
>
> Q: Motivation for the practical applications
> A: Following reviewers’ suggestions, we downplay our contribution regarding real-world applications in this revision. Instead, we focus on the in-depth analysis of our core algorithm under different settings. We present the domain adaptation and data poisoning attack as two of these settings (i.e., different initial weight distribution and learning objective) rather than ready-to-use applications. We also added a few strong baselines for each setting. These settings help us gain a deeper understanding of the application scopes and limitations of our current method.
>
> Q: Analysis of the proposed procedure; which Loss it minimizes and if it indeed guaranteed to converge to the minimum of that loss.
> A: For the general case, the loss minimized is the task loss using the **updated** weights. E.g., classification loss of weights after gradient updates using distilled data. In the setting data-poisoning malicious objective, the loss is the classification loss w.r.t. modified labels (i.e., labels with attack category changed to target category). Similar to standard neural network training, there is no theoretical guarantee yet that updated weights are at a (local) minimum. However, empirical results in Sec. 4 show high test accuracy and thus demonstrate that dataset knowledge is indeed distilled into the trained data.
>
> In the linear regression case analysis in Sec. 3.3, the loss is the MSE loss function \ell(d, t, theta) = 1/(2N) || d \theta - t ||^2, where d is the data matrix, t is the target vector, and \theta is the weight vector. For such models, there always exists distilled data that can train arbitrary initial weights to optimum. One choice is to pick any global minimum \theta*, and choose distilled \tilde{d}= N * I and \tilde{t} = N * \theta*. Sec. 3.3 investigates the minimum number of distilled data required. We find that at least D samples are needed, where D is the number of feature dimensions. We have updated Sec. 3.3 and clarified this point.
>
> Q: Importance of learning with few gradient steps
> A: The number of gradient steps is tightly related to the number of distilled images. Often times, we only need to train a model with fewer steps on smaller datasets.  We have revised Sec. 1 to mention few gradient steps as a byproduct of our method, instead of a major motivation.
>
>
>
> [1] Motiian et al., "Few-shot adversarial domain adaptation." NeurIPS 2017.
>
>
> edit: fix a typo, update NeurIPS acronym

---

### Official Review · AnonReviewer3 · 2018-10-31
**Original problem and well written paper, but that lacks comparisons to baselines**

**Rating:** 6
**Confidence:** 4

**Review:**


=== Post rebuttal update ===

Thanks to the addition of better baselines, I've increased my score for this paper. While I'm still not super convinced of its potential for application, I find the idea original and worth discussing at the conference.

=== Pre-rebuttal review ===
This paper presents an approach to compress a dataset into a much smaller number of synthetic samples that are optimized to yield as good performance as possible when a given model is trained on that smaller dataset. This is done by unrolling the gradient descent procedure of training such a model to allow for gradient-based optimization of synthetic samples themselves as well as the used learning rates.

In summary, my evaluation is as follow:

*Pros*
- Pretty original problem formulation
- Generally well written paper

*Cons*
- Lack of comparison with simple baselines in basic dataset distillation setting
- Use in practical applications (domain adaptation, data poisoning) yet to be convincingly demonstrated
- Possibly a mistake in the theoretical analysis of the linear case

Indeed, I found the paper to be generally quite clear and enjoyed reading it. One minor thing I struggled a bit with is the distinction between "SG steps" and "Epochs" (I believe the former corresponds to when the synthetic samples are different between GD steps, whereas the later corresponds to the number of times the method repeatedly cycles over these samples) so I would perhaps encourage the authors to emphasize that difference.

I also find the problem statement that proposed to be interesting and thought provoking, and the solution that's proposed seems quite appropriate and well thought out.

That said, I'm worried about the following:

- Unless I misunderstood, in the basic dataset distillation setting a comparison is never provided with training on a randomly selected subset of the training set. Presumably the results are worse, but I think these results should be in the paper. I would also argue for having another baseline, which would try to (approximately) optimize the choice of which training examples are put in the subset. A very simple approach would be to take the 200 runs already performed for the random selection and select the subset providing the best accuracy on the full training set and only report the performance of that subset (instead of the mean and std of all 200 runs). In short, this would help determine to what extent there is value to synthesizing entirely new samples. Moreover, I think a simple alternative baseline for creating synthesized samples should be considered. Specifically, I'd personally would like to know the performance of using per-class k-means clustering and training on the cluster centroids as the distilled dataset.

- While I appreciate that the authors identify potential applications and report some results on them, I think they currently fall short of convincing the reader of the potential of dataset distillation for these applications. For domain adaptation, no actual domain adaptation baseline is compared against (a good candidate would be method from Daume III 2007, at the very least). For data poisoning, I find that the assumptions for attacks are pretty strong, i.e. that a) you have access to the parameters of the pre-trained models to attack and b) that the model is doing additional updates *only* on the synthesized data. If there are reasons to think that such assumptions are reasonable, I'd at least expect the paper to motivate why that is.

- In the analysis of the simple linear case, in Equation 7, there appears to be a mistake, specifically some missing parentheses:

d^Td( (I-\eta/M d^Td)\theta_0 + \eta/M \tilde{d}^T\tilde{t}) = d^T t

i.e. there should be parentheses right after "d^Td" and right before "=". This is from replacing \theta^* by the expression for \theta_1 in Equation 6, which is what I think Equation 7 is supposed to be doing. This possibly doesn't affect some of the conclusions taken from this section, but I'd like to see this potential mistake discussed/addressed.


That said, if the authors can sufficiently address the 3 points above, I'd be willing to increase my rating for this paper.

Finally, I have a few other more minor (nice-to-have) points:
- Having in the related work a discussion on the relationship with coreset methods would be nice
- Experiments showing how well the distilled datasets transfer to different network architectures than those used in training would be interesting? Even other ML algorithms would be quite interesting?
- "We often find that the number of distilled images required to achieve good performance is an informative indicator of the dataset diversity" => I'm not sure what in the paper actually justifies / demonstrates this statement.
- Figure 4 is presented as an "Ablation study", but an ablation study is where you remove certains parts of a model or algorithm and see what happens, which isn't the case here. I think it's better described as a hyper-parameter sensitivity study.
- Some typos:
   * the below objective => the objective below
   * w.r.t. to => w.r.t
   * the discrete part rather => the discrete parts rather
   * necesary => necessary

---

> ### Author Response · Authors · 2018-11-27
> **Re:  Original problem and well written paper, but that lacks comparisons to baselines**
>
> Q: More baselines in basic dataset distillation setting
> A: Following your suggestions, we have included many more baselines in this revision, including using data from random real images, optimized real images (top 20% best-performing random set of real images over 50 runs), average real images, and k-means cluster centroids. Additionally, for the basic data distillation setting, we also included comparisons against k-NN. Our method with fixed initialization beats all the baselines on MNIST and CIFAR10. Other than k-means with k-NN on MNIST, our method with random unknown initialization beats all the other baselines on MNIST and CIFAR10. We have added these experiments to Sec. 4.1.
>
> Q: Better baselines in domain adaptation (pre-trained initial weights) setting
> A: For the pre-trained initial weights setting (domain adaptation), we now compare our method against a SOTA few-shot domain adaptation method [1]. Additionally, to show our results on larger datasets, we added a new experiment of adapting an AlexNet pre-trained on ImageNet to PASCAL-VOC and CUB-200. Our method significantly outperforms all the baselines (47% better on PASCAL-VOC and 31% better on CUB-200). Sec. 4.2 in the revision includes these new experiments.
>
> Q: Limitations of the data poisoning attack (pre-trained initial weights + malicious objective function) setting
> A: Our method does **not** require access to the exact model weights. The distilled images are trained on many well-trained classifiers (2000 classifiers in our experiments) and thus can generalize to unseen models. In the paper, the results are evaluated on 200 held-out models. We clarified this better in Sec. 4 paragraph 1 and Sec. 4.2.
>
> Q: Typo in linear analysis
> A: Thanks for pointing this out. We have corrected it in Sec. 3.3 of this revision.
>
> Q: Related work on core-set
> A: Thanks for your suggestions. We have added a new related work section on core-set and instance selection in Sec. 2.
>
> Q: GD Steps and Epochs
> A: Each step is associated with a different batch of distilled data. All steps are sequentially cycled over for #epochs times. We clarified this in Sec. 3.4.
>
> Q: Generalize to other architectures and algorithms
> A: These are both potentially promising directions to explore. However, it might require training models on stochastic architectures with stochastic initializations, which is beyond our current computational resources. We leave it as future work.
>
> Q: Relation between #Images and dataset diversity
> A: We agree that our experiment paper has not fully justified the claimed correlation. We have removed the sentence in Sec. 2.
>
> Q: Figure 4 title
> A: Following your suggestion, we renamed Figure 4 to “hyperparameter sensitivity studies”.
>
> Q: Other typos
> A: Thanks for your detailed comments! We have fixed them in this revision.
>
>
> [1] Motiian et al., "Few-shot adversarial domain adaptation." NeurIPS 2017.
>
> edit: fix a typo, update NeurIPS acronym

---

### Official Review · AnonReviewer1 · 2018-11-02
**An algorithm to reduce datatset size for  NNs**

**Rating:** 5
**Confidence:** 4

**Review:**

The paper presents an algorithm for compressing the size of entire training data into a few synthetic training samples. The method is based on neural networks and is applied on image datsets. The authors comment two possible applications of their method domain adaptation and effective data poisoning attack.

The proposed technique seems to be limited to neural networks since it seems that is linked to the initialization of the networks. In this aspect, it could be interesting to have a more general method.

There are related works that are not commented, for instance :

Olvera-López, J. Arturo, et al. "A review of instance selection methods." Artificial Intelligence Review 34.2 (2010): 133-143.


Experimental section is weak. Few datasets are considered, other problems should be added. Additionally, related methods should be included to  compare the performance of the proposal. Some comments about the computational cost should be inserted. In this aspect, the experimental section should be improved following these recommendations.

---

> ### Author Response · Authors · 2018-11-27
> **Re: An algorithm to reduce datatset size for NNs**
>
> Q: Related work on instance selection
> A: Thank you for your reference. In Sec. 2 of this revision, we have added discussions on core-set construction and instance selection.
>
> Q: More datasets and related  baselines in the experiment section
> A: In this revision, we have added substantially more and stronger baselines for all the experiment settings, including k-means centroids, optimized set of real images, and a SOTA few-shot domain adaptation method [1]. Additionally, we have included a new experiment in adapting an AlexNet pre-trained on ImageNet to PASCAL-VOC and CUB-200 classification, two commonly used large-scale image datasets. Our method beats the baselines and a recent domain adaptation method [1] in most settings and datasets. Please see Sec. 4.1 and Sec. 4.2 for more details.
>
> Q: Computational cost
> A: During the evaluation, our method is as fast as a few standard gradient descent steps. For training, we run our experiments on NVIDIA Titan Xp and V100 GPUs. We use one GPU for fixed initialization, and four GPUs for random unknown initial weights. Training typically takes 1 to 4 hours. We have added detailed running time information in Sec. 4.
>
>
> [1] Motiian et al., "Few-shot adversarial domain adaptation." NeurIPS 2017.
>
>
> edit: fix a typo, update NeurIPS acronym

---

### Public Comment · (anonymous) · 2018-10-05
**Question about hyperparameters**

I'm trying to reproduce your results in Sec. 4, but had some questions about the hyperparameters in Algorithm 1:

1) What is a reasonable size for the minibatch of real samples (i.e., 'n' on line 3)?
2) What is a reasonable number of \theta_0 samples (i.e., how many time is the loop on line 5 run)?
3) What is a reasonable number of outer optimization steps (i.e., 'T' on line 2)?

Thanks!

---

> ### Author Response · Authors · 2018-10-05
> **Re: Question about hyperparameters**
>
> Hi,
>
> Thanks for your interest in our paper! I will reply to your questions below:
>
> 1) What is a reasonable size for the minibatch of real samples (i.e., 'n' on line 3)?
>
> We used minibatch of 1024 real samples to generate the results in paper. We also tried 256 and 512 some time when working on the project. It didn't seem to make much difference.
>
> 2) What is a reasonable number of \theta_0 samples (i.e., how many time is the loop on line 5 run)?
>
> In the experiments, we used 4~16 samples per iteration, depending on the distributed training set-up. In my experience, even 1 sample per iteration can yield reasonable results.
>
> 3) What is a reasonable number of outer optimization steps (i.e., 'T' on line 2)?
>
> For datasets other than USPS, we iterate over the training set 150 times. for USPS, we iterate 200 times.
>
> Hope that this helps!

---

> > ### Public Comment · (anonymous) · 2018-10-05
> > **Few more questions**
> >
> > This helps - thanks! Just a couple more clarifications -- what do you use for \alpha and \eta_0?

---

> > > ### Author Response · Authors · 2018-10-05
> > > **Re: Few more questions**
> > >
> > > In most of our experiments, we initialize the \eta for all steps to be `0.001, and use \alpha with initial value 0.01 and exponentially decay with a factor of 0.5 every 23 epochs (sweeps over the training dataset) for USPS and 30 epochs for other datasets. The fast data-poisoning attack experiments use uses initial \eta 0.02 and initial \alpha 0.02.
> > >
> > > There is no particular reason about the difference. In my experience, reasonable values about that range work fine.
> > >
> > > I plan to clean-up and release our code when it is ready. You are more than welcome to try it out when that happens! Unfortunately, for anonymity reasons, this may take a while.

---

### Author Response · Authors · 2018-11-27
**Summary of changes**

We thank all the reviewers for their helpful comments. We are glad that they found the problem original, the paper well-written, and the method well-thought out. We have addressed individual questions raised by the reviewers in separate posts.

In the new draft of the paper, we significantly improve the experiments and add several strong baselines suggested by reviewers. We agree with reviewers to “go deeper rather than wider”. Therefore, in the new draft, we downplay the applications and focus on the extensive analysis and different settings of our core algorithm. Below we summarize the major changes.

1. Improved experiments section and added strong baselines (Sec. 4):
    + Sec 4.1 and 4.2: We compared our method against various baselines for the basic dataset distillation setting, including k-means images, random real images, and optimized real images over 50 runs, suggested by R3. We also compare our method with these new baselines in the adaptation and malicious attack settings. For adaptation, we added a comparison against the SOTA few-shot domain adaptation method [1]. Our method beats the baselines and the few-shot domain adaptation method [1] in most settings and datasets. (R1, R2, R3)

    + Sec 4.2: We added experimental results for adaptation from AlexNet pre-trained on ImageNet to PASCAL-VOC and CUB-200, two commonly used large-scale computer vision datasets. Our method significantly outperforms all the baselines (47% better on PASCAL-VOC and 31% better on CUB-200). (R1, R2, R3)

    + Sec. 4: We added computational cost and running time information about our training. (R2)

2. Sec. 3: We now present domain adaptation and data-poisoning attacks as different dataset distillation settings with different initial weight distributions (e.g., fixed or random pre-trained weights) and different final objectives (classification, attack), instead of ready-to-use applications. We leave the potential applications as future work.  (R2, R3)

3. Sec. 2: We added a related work section on core-set construction, instance selection, and classical linear algorithms, and extended an existing section to discuss relations with teaching dimension. (R1, R2, R3)

4. Sec. 3: We fixed minor typos, including the one in the linear case analysis. (R3)

5. Sec. 3.3: We clarified that for the linear case, there always exists distilled data that can train arbitrary initial weights to optimum, although this section focuses on the lower bound on the size of such data. (R2)


[1] Motiian, Saeid, et al. "Few-shot adversarial domain adaptation." NeurIPS 2017.

edit: fix a typo, update NeurIPS acronym

---

### Meta-Review · Area_Chair1 · 2018-12-14
**Original idea, but lacks a practical use case.**

**Confidence:** 3
**Recommendation:** Reject

**Metareview:**

The reviewers agree that the idea for dataset distillation is novel, however it is unclear how practical it can be. The paper has been significantly improved through the addition of new baselines, however ultimately the performance is not quite good enough for the reviewers to advocate strongly on its behalf. Perhaps the paper would be better motivated by finding a realistic scenario in which it would make sense for someone to use this approach over reasonable alternatives.